# It begins with a boundary: A geometric view on probabilistically robust learning

## Abstract

Although deep neural networks have achieved super-human performance on many classification tasks, they often exhibit a worrying lack of robustness towards adversarially generated examples. Thus, considerable effort has been invested into reformulating Empirical Risk Minimization (ERM) into an adversarially robust framework. Recently, attention has shifted towards approaches which interpolate between the robustness offered by adversarial training and the higher clean accuracy and faster training times of ERM. In this paper, we take a fresh and geometric view on one such method—Probabilistically Robust Learning (PRL) [Robey et al., 2022]. We propose a geometric framework for understanding PRL, which allows us to identify a subtle flaw in its original formulation and to introduce a family of probabilistic nonlocal perimeter functionals to address this. We prove existence of solutions using novel relaxation methods and study properties as well as local limits of the introduced perimeters.

## 1  Introduction

The fragility of DNN-based classifiers in the face of adversarial examples [Goodfellow et al., 2014, Chen et al., 2017, Qin et al., 2019, Cai et al., 2021] and distributional shifts [Quinoñero Candela et al., 2008, Hendrycks et al., 2021] is by now nearly as familiar as their successes. In light of this, a multitude of works (see Section 1.4) propose replacing standard Empirical Risk Minimization (ERM) [Vapnik, 1999] with a more robust alternative (see, e.g., Madry et al. [2017]). Unfortunately there is no free lunch: robust classifiers frequently exhibit degraded performance on clean data and significantly longer training times [Tsipras et al., 2018]. Consequently, identifying frameworks which balance performance and robustness is of pressing interest to the Machine Learning (ML) community, and over the past several years many such frameworks have been proposed [Zhang et al., 2019, Wang et al., 2020, Robey et al., 2022]. Moreover, it is crucial that the mechanism by which such frameworks balance these competing aims be understood.

Beginning with the Probabilistically Robust Learning (PRL) of Robey et al. [2022] we analyze such frameworks geometrically. This perspective reveals a subtle, paradoxical aspect of PRL: sometimes the adversary modeled by this framework corrects, instead of exploits, the learner! Fortunately, the geometric perspective we propose suggests a natural remedy which leads to an interpretation of the corrected PRL as regularized ERM where a certain nonlocal notion of length (or perimeter) of the decision boundary acts as a regularizer. We exemplify this correction in Figure 1. The interpretation of PRL as perimeter-regularized ERM leads us to further generalizations, and we provide a novel view of the Conditional Value at Risk (CVaR) relaxation of PRL proposed by Robey et al. [2022].

Submitted to 37th Conference on Neural Information Processing Systems (NeurIPS 2023). Do not distribute.

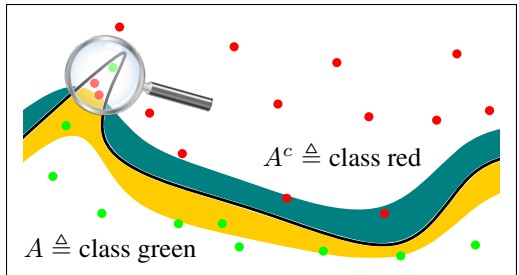 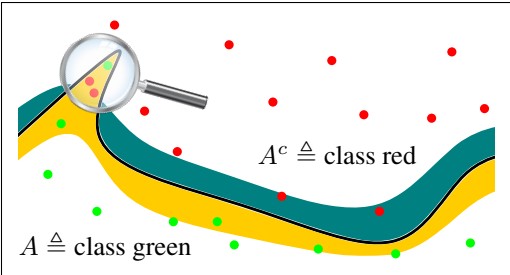

(a) Robey et al. [2022]: The probabilistically non-robust region (**magnified**) reduces the loss.

(b) Our model: The probabilistically non-robust region is correctly identified and penalized.

Figure 1: Penalization effect of the original model [Robey et al., 2022] (**left**) and ours (**right**): The solid black is the decision boundary of a non-robust classifier induced by the set $A$. Both models penalize the numbers of green points in the yellow region and red points in the teal region. However, the original model *favors non-robust regions* of $A$ for which most perturbations correct the class. Our model identifies this region as non-robust and penalizes it accordingly.

## 1.1    From empirical risk minimization to robustness

Given an input space $\mathcal{X}$, an output space $\mathcal{Y}$, a probability measure $\mu \in \mathcal{P}(\mathcal{X} \times \mathcal{Y})$, a loss function $\ell : \mathcal{Y} \times \mathcal{Y} \to \mathbb{R}$, and a hypothesis class $\mathcal{H}$, the standard risk minimization problem is

$$\inf_{h \in \mathcal{H}} \mathbb{E}_{(x,y) \sim \mu} \left[ \ell(h(x), y) \right]. \tag{1}$$

For training classifiers which are robust against adversarial attacks Goodfellow et al. [2014], Madry et al. [2017] suggested adversarial training:

$$\inf_{h \in \mathcal{H}} \mathbb{E}_{(x,y) \sim \mu} \left[ \sup_{x' \in B_\varepsilon(x)} \ell(h(x'), y) \right]. \tag{2}$$

Here $\mathcal{X}$ is assumed to have the structure of a metric space and $B_\varepsilon(x)$ for $\varepsilon \geq 0$ denotes the (open or closed) ball of radius $\varepsilon$ around $x$.

The recent work by Robey et al. [2022] offered an alternative to adversarial training in order to reduce the (in general) large trade-off between accuracy and robustness inherent in (2), see Tsipras et al. [2018], Robey et al. [2022] for discussion. Instead of requiring classifiers to be robust to *all* available attacks around a point $x$—as enforced through the supremum in (2)—one may consider a less stringent notion of robustness, only requiring classifiers to be robust to $100 \times (1 - p)\%$ of possible attacks when attacks are drawn from a certain distribution $\mathfrak{p}_x$ centered at $x$. For this, the authors introduced the so-called $p$-ess sup operator for $p \in [0, 1)$ and suggested replacing (2) by

$$\inf_{h \in \mathcal{H}} \mathbb{E}_{(x,y) \sim \mu} \left[ p\text{-} \operatorname*{ess\,sup}_{x' \sim \mathfrak{p}_x} \ell(h(x'), y) \right], \tag{3}$$

where $\{\mathfrak{p}_x\}_{x \in \mathcal{X}}$ is a family of probability distributions. The prototypical example to keep in mind for $\mathcal{X} = \mathbb{R}^d$ is the uniform distribution over the $\varepsilon$-ball around $x$, i.e., $\mathfrak{p}_x := \mathrm{Unif}(B_\varepsilon(x))$, which is particularly relevant when dealing with adversarial attacks on image classifiers.

For a probability distribution $\mathfrak{p}$ and a function $f$, the quantity $p\text{-} \operatorname*{ess\,sup}_{x' \sim \mathfrak{p}} f(x')$ is defined as the smallest value $t \in \mathbb{R}$ such that the probability of a randomly chosen point $x' \sim \mathfrak{p}$ satisfying $f(x') > t$ is smaller than $p$, which reduces to the usual essential supremum of $f$ with respect to $\mathfrak{p}$ if $p = 0$:

$$p\text{-} \operatorname*{ess\,sup}_{x' \sim \mathfrak{p}} f(x') := \inf \left\{ t \in \mathbb{R} \, : \, \mathbb{P}_{x' \sim \mathfrak{p}} \left[ f(x') > t \right] \leq p \right\}.$$

To better understand the model (3) we temporarily restrict our attention to binary classification (i.e., $\mathcal{Y} = \{0, 1\}$) using indicator functions of admissible sets (i.e., $\mathcal{H} := \{\mathbf{1}_A \, : \, A \in \mathcal{A}\}$). Note that we identify the two expressions $\mathbf{1}_A(x) = \mathbf{1}_{x \in A}$. We focus on the 0-1 loss $\ell(\tilde{y}, y) = \mathbf{1}_{\tilde{y} \neq y}$ which equals one if $y \neq \tilde{y}$ and zero otherwise. In this scenario (1) reduces to the geometric problem

$$\inf_{A \in \mathcal{A}} \left\{ \mathrm{R}_{\mathrm{std}}(A) := \mathbb{E}_{(x,y) \sim \mu} \left[ y \mathbf{1}_{x \in A^c} + (1 - y) \mathbf{1}_{x \in A} \right] \right\}, \tag{4}$$

and minimizers are called Bayes classifiers. Similarly, adversarial training (2) can be rewritten as

$$\inf_{A \in \mathcal{A}} \left\{ \mathrm{R}_{\mathrm{adv}}(A) := \mathbb{E}_{(x,y) \sim \mu} \left[ y \mathbf{1}_{x \in (A^c)^{\oplus \varepsilon}} + (1-y) \mathbf{1}_{x \in A^{\oplus \varepsilon}} \right] \right\}, \tag{5}$$

where for a set $A \in \mathcal{A}$ its fattening by $\varepsilon$-balls is defined as $A^{\oplus \varepsilon} := \bigcup_{x \in A} B_\varepsilon(x)$. Hence (5) enforces that all points with distance at most $\varepsilon$ to the decision boundary be adversarially robust.

On the other hand the PRL model (3) reduces to

$$\inf_{A \in \mathcal{A}} \left\{ \mathrm{R}_{\mathrm{prob}}(A) := \mathbb{E}_{(x,y) \sim \mu} \left[ y \mathbf{1}_{\mathbb{P}_{x' \sim \mathfrak{p}_x}[x' \in A^c] > p} + (1-y) \mathbf{1}_{\mathbb{P}_{x' \sim \mathfrak{p}_x}[x' \in A] > p} \right] \right\}, \tag{6}$$

where $A^{\oplus \varepsilon}$ is replaced by a "probabilistic fattening", i.e., one considers the set of all $x$ for which the probability that a neighboring point sampled from $\mathfrak{p}_x$ lies inside $A$ is larger than $p$. To the best of our knowledge, existence of solutions for (6) or even (3) has not been proved so far.

## 1.2 Geometric modification of probabilistically robust learning

To motivate our geometric modification of the PRL model from Robey et al. [2022], it is insightful to investigate the regularization effect that PRL has compared to standard risk minimization. We let $\rho_i(\bullet) := \mu(\bullet \times \{i\})$ denote the non-normalized conditional distributions of the points with label $i$. Subtracting the standard risk in (4) from the one in (6) and disintegrating using $\rho_0$ and $\rho_1$ we obtain

$$\mathrm{R}_{\mathrm{prob}}(A) - \mathrm{R}_{\mathrm{std}}(A)$$
$$= \int_{\mathcal{X}} \mathbf{1}_{\mathbb{P}_{x' \sim \mathfrak{p}_x}[x' \in A] > p} - \mathbf{1}_{x \in A} \, \mathrm{d}\rho_0(x) + \int_{\mathcal{X}} \mathbf{1}_{\mathbb{P}_{x' \sim \mathfrak{p}_x}[x' \in A^c] > p} - \mathbf{1}_{x \in A^c} \, \mathrm{d}\rho_1(x). \tag{7}$$

We highlight that this expression *does not constitute a non-negative functional of $A$*. Hence the loss function in (6) is not a regularized version of the standard risk (4) and in fact can be strictly smaller. This observation reveals a subtle flaw in the approach of Robey et al. [2022]: Points which lie in thin or spike-like regions of $A$ penetrating the other class and that are more likely to have the label zero than the label one (meaning they lie in the set $\{\rho_0 > \rho_1\}$) yield negative contributions in (7) and are hence *favored*. Such a scenario is visualized on the left side of Figure 1. From an adversarial perspective this means that points which are already misclassified are attacked nevertheless, which can lead to the bizarre situation that the adversary helps the learner by putting these points in the correct class with high probability, thereby reducing both adversarial robustness and clean accuracy.

We fix this by designing a probabilistically robust risk as non-negative regularization of the standard risk. For this we define probabilistic perimeter functionals which only penalize points which are classified correctly *and* admit a large portion of attacks around them, see the right side of Figure 1.

## 1.3 Our contributions

Our main contributions are the following:

- We address the geometric limitation of the model by Robey et al. [2022] by introducing a family of perimeter regularizations.
- We prove existence of soft and hard binary classifiers under weak conditions on the family of perimeters and hypothesis classes, using novel relaxation techniques.
- We investigate the relationship between the introduced family of perimeters and local perimeters in Euclidean space for small adversarial budgets.
- We extend our models to encompass general loss functions and hypothesis classes. Our numerical experiments demonstrate that our geometric correction can enhance the adversarial robustness of probabilistically robust classifiers without compromising clean accuracy.

## 1.4 Related work

Adversarial training was developed by Goodfellow et al. [2014], Madry et al. [2017] as an approach to train networks that are less sensitive to adversarial attacks. Shafahi et al. [2019] reduced its computational complexity by reusing gradients from the backpropagation when training neural

networks. Wong et al. [2020] showed that training with noise perturbations followed by a single signed gradient ascent (FGSM) step can be on par with adversarial training while being much cheaper. This approach was picked up and improved upon by Andriushchenko and Flammarion [2020] based on gradient alignment. Different authors also investigated test-time robustification of pretrained classifiers using randomized smoothing [Cohen et al., 2019] or geometric / gradient-based approaches [Schwinn et al., 2021, 2022]. While some of the previous models use a combination of random perturbations and gradient-based adversarial attacks to robustify classifiers, Robey et al. [2022] proposed probabilistically robust learning, which is entirely based on random perturbations. PRL aims to interpolate between clean and adversarial accuracy and enjoys the favorable sample complexity of vanilla empirical risk minimization; see also Raman et al. [2023] for more insights on this issue. Connections between adversarial training and local perimeter regularization of decision boundaries were explored by García Trillos and Murray [2022] and then rigorously tied by Bungert and Stinson [2022]. Our work is in line with a series of papers [Pydi and Jog, 2021, Awasthi et al., 2021a,b, Frank and Niles-Weed, 2022, Frank, 2022, Bungert et al., 2023, García Trillos et al., 2023] that explore the existence of solutions to adversarial training problems in different settings. These existence proofs involve dealing with different kinds of measurability issues, depending on whether open or closed balls $B_\varepsilon(x)$ are used in the attack model. For open balls one can work with the Borel $\sigma$-algebra $\mathcal{A} = \mathfrak{B}(\mathcal{X})$ [Bungert et al., 2023], whereas closed balls require the use of the universal $\sigma$-algebra to make sure that $A^{\oplus\varepsilon}$ is measurable [Pydi and Jog, 2021, Awasthi et al., 2021a,b]. Recently, these results were improved by García Trillos et al. [2023] who also proved for the case of multi-class classification that even for the closed ball model Borel measurable classifiers (albeit not necessarily indicator functions of measurable sets) exist and that for all but countably many values of the adversarial budget $\varepsilon > 0$ the open and the closed ball models have the same minimal value.

## 2 Geometry and existence of probabilistically robust classifiers

### 2.1 The binary classification setting with 0-1 loss

In this section we shall introduce our baseline model, which is based on a suitable geometric regularization of the standard risk. Later we shall embed it into a family of models. For clarity we first discuss hard classifiers (characteristic functions of sets) and then soft classifiers (functions with values in $[0, 1]$). The generalization to general models and loss functions is postponed to Section 3.

We start by defining the *probabilistic perimeter* for $p \in [0, 1)$ of an admissible set $A \in \mathcal{A}$ as follows:

$$
\begin{aligned}
\mathrm{ProbPer}(A) := \ &\rho_0\left(\{x \in A^c \,:\, \mathbb{P}_{x'\sim\mathfrak{p}_x}[x' \in A] > p\}\right) \\
&+ \rho_1\left(\{x \in A \,:\, \mathbb{P}_{x'\sim\mathfrak{p}_x}[x' \in A^c] > p\}\right).
\end{aligned}
\tag{8}
$$

$\mathrm{ProbPer}(A)$ penalizes correctly classified points $x$ for which more than $100 \times p\,\%$ of their neighbors, sampled from $\mathfrak{p}_x$, constitute an attack. The perimeter can be rewritten in integral form:

$$
\begin{aligned}
\mathrm{ProbPer}(A) = &\int_{\mathcal{X}} \mathbf{1}_{x \in A \,\vee\, \mathbb{P}_{x'\sim\mathfrak{p}_x}[x'\in A] > p} - \mathbf{1}_{x \in A} \,\mathrm{d}\rho_0(x) \\
&+ \int_{\mathcal{X}} \mathbf{1}_{x \in A^c \,\vee\, \mathbb{P}_{x'\sim\mathfrak{p}_x}[x'\in A^c] > p} - \mathbf{1}_{x \in A^c} \,\mathrm{d}\rho_1(x) \\
= &\int_{\mathcal{X}} \mathbf{1}_{x \in A^c} \mathbf{1}_{\mathbb{P}_{x'\sim\mathfrak{p}_x}[x'\in A] > p} \,\mathrm{d}\rho_0(x) + \int_{\mathcal{X}} \mathbf{1}_{x \in A} \mathbf{1}_{\mathbb{P}_{x'\sim\mathfrak{p}_x}[x'\in A^c] > p} \,\mathrm{d}\rho_1(x).
\end{aligned}
\tag{9}
\tag{10}
$$

The first reformulation (9) should be compared to (7), while the one in (10) will be useful later on. The use of the term perimeter to describe the functional $\mathrm{ProbPer}$ will become more apparent shortly in Section 2.4, and at this point it is worth highlighting that $\mathrm{ProbPer}$ is always a non-negative quantity. This motivates introducing the following regularized risk

$$
\mathrm{ProbR}(A) := \mathrm{R}_{\mathrm{std}}(A) + \mathrm{ProbPer}(A), \qquad A \in \mathcal{A}.
\tag{11}
$$

Our first theorem states that $\mathrm{ProbR}$ equals the expected maximum of the sample-wise standard risk and the probabilistically robust risk from Robey et al. [2022], cf. (4) and (6).

**Theorem 1.** *For all $A \in \mathcal{A}$ it holds that*

$$
\mathrm{ProbR}(A) = \mathbb{E}_{(x,y)\sim\mu}\left[\max\left\{\mathbf{1}_{\mathbb{P}_{x'\sim\mathfrak{p}_x}[\mathbf{1}_A(x')\neq y] > p}, \mathbf{1}_{\mathbf{1}_A(x)\neq y}\right\}\right].
\tag{12}
$$

The interpretation of the statement of this theorem in the light of Figure 1 is clear: Only if a point $x$ is correctly classified—meaning $\mathbf{1}_{\mathbf{1}_A(x) \neq y} = 0$—the probabilistically robust regularization kicks in through the first term in the maximum. Points which are incorrectly classified will always be penalized even if most attacks correct the label, i.e., if $\mathbf{1}_{\mathbb{P}_{x' \sim \mathfrak{p}_x}[\mathbf{1}_A(x') \neq y] > p} = 0$. Thus, minimizing ProbR instead of $\mathrm{R}_{\mathrm{prob}}$ corrects the pathology identified in Section 1.2.

## 2.2 Extensions in the binary classification setting

Given the formula of ProbPer in (10), several natural extensions suggest themselves. E.g., one may replace the indicator function $\mathbf{1}_{t > p}$ with a different function $\Psi(t)$ to define other notions of *perimeter*

$$\mathrm{ProbPer}_\Psi(A) := \int_{\mathcal{X}} \mathbf{1}_{x \in A^c} \Psi\left(\mathbb{P}_{x' \sim \mathfrak{p}_x}\left[x' \in A\right]\right) \, \mathrm{d}\rho_0(x) \\ + \int_{\mathcal{X}} \mathbf{1}_{x \in A} \Psi\left(\mathbb{P}_{x' \sim \mathfrak{p}_x}\left[x' \in A^c\right]\right) \, \mathrm{d}\rho_1(x) \tag{13}$$

as well as their corresponding probabilistically robust losses

$$\mathrm{ProbR}_\Psi(A) := \mathrm{R}_{\mathrm{std}}(A) + \mathrm{ProbPer}_\Psi(A). \tag{14}$$

For $\Psi(t) := \mathbf{1}_{t > p}$ the perimeter $\mathrm{ProbPer}_\Psi$ reduces to ProbPer and so do the associated risks. Of particular interest is $\Psi_p(t) := \min\{t/p, 1\}$—the smallest concave function that lies above $\Psi(t) = \mathbf{1}_{t > p}$—which will allow us to develop deep connections between the theoretical and computational aspects of probabilistically robust learning. Our relaxation using the function $\Psi$ is very similar to the one by Raman et al. [2023] who proved PAC learnability if $\Psi$ is Lipschitz, see Appendix A.6 for more details. In order to rigorously study $\mathrm{ProbR}_\Psi$ we first make our setting precise.

**Assumption 1.** We let $\mathcal{X}$ be a set and $\mathcal{A} \subset 2^{\mathcal{X}}$ be a $\sigma$-algebra. We assume that:

- $(\mathcal{X} \times \mathcal{Y}, \mathcal{A} \otimes 2^{\{0,1\}}, \mu)$ is a probability space;

- $(\mathcal{X}, \mathcal{A}, \rho)$ is a probability space, where we define $\rho(\bullet) := \mu(\bullet \times \{0, 1\})$;

- $\{\mathfrak{p}_x\}_{x \in \mathcal{X}}$ is a family such that $(\mathcal{X}, \mathcal{A}, \mathfrak{p}_x)$ is a probability space for $\rho$-almost every $x \in \mathcal{X}$.

The following theorem establishes existence of minimizers of the risk $\mathrm{ProbR}_\Psi$ for concave and non-decreasing functions $\Psi$. This existence result is astonishing since the standard method of calculus of variations is not directly applicable, with the reason being that problem (15) does not provide enough compactness for lower semicontinuity of the perimeter functional $\mathrm{ProbPer}_\Psi$. Instead, the proof is based on convex relaxations to soft classifiers where we use a lower semicontinuous surrogate functional and a total variation defined through a coarea formula which—if $\Psi$ is concave and non-decreasing—lower-bounds the surrogate.

**Theorem 2.** *Suppose* $\Psi : [0, 1] \to [0, 1]$ *is concave and non-decreasing, and that Assumption 1 holds. Then, there exists a solution to the problem*

$$\inf_{A \in \mathcal{A}} \mathrm{ProbR}_\Psi(A). \tag{15}$$

Furthermore, $\mathrm{ProbR}_\Psi$ can also be interpreted as a sample-wise maximum, analogous to Theorem 1.

**Theorem 3.** *For all* $A \in \mathcal{A}$ *and measurable* $\Psi : [0, 1] \to [0, 1]$ *it holds*

$$\mathrm{ProbR}_\Psi(A) = \mathrm{R}_{\mathrm{std}}(A) + \mathrm{ProbPer}_\Psi(A) \\ = \mathbb{E}_{(x,y) \sim \mu}\left[\max\left\{\Psi\left(\mathbb{P}_{x' \sim \mathfrak{p}_x}\left[\mathbf{1}_A(x') \neq y\right]\right), \mathbf{1}_{\mathbf{1}_A(x) \neq y}\right\}\right].$$

Note that for the non-concave function $\Psi(t) = \mathbf{1}_{t > p}$ an existence proof along the lines of Theorem 2 is not available since certain relaxation techniques therein rely on concavity of $\Psi$. However, in the next section we shall provide an existence theorem for soft classifiers which is valid for very general functions $\Psi$, including $\Psi(t) = \mathbf{1}_{t > p}$.

## 2.3 Extension to soft classifiers

Another natural extension features "soft classifiers" instead of indicator functions of admissible sets. Such classifiers are particularly relevant since they include the neural network based models with `Softmax` activation in the last layer which are used in practice. We start by defining a suitable regularization functional for soft classifiers. Given a $\mathcal{A}$-measurable function $u : \mathcal{X} \to [0,1]$ we define

$$J_\Psi(u) := \int_{\mathcal{X}} (1 - u(x)) \, \Psi\left(\mathbb{E}_{x' \sim \mathfrak{p}_x} [u(x')]\right) \, \mathrm{d}\rho_0(x)$$
$$+ \int_{\mathcal{X}} u(x) \Psi\left(\mathbb{E}_{x' \sim \mathfrak{p}_x} [1 - u(x')]\right) \, \mathrm{d}\rho_1(x) \tag{16}$$

which satisfies $J_\Psi(\mathbf{1}_A) = \mathrm{ProbPer}_\Psi(A)$ for every choice of $\Psi$. Hence, it is a natural generalization of the perimeter to soft classifiers and one could call $J_\Psi$ a total variation. However, it is neither positively homogeneous nor convex so this name would be misleading. Instead, for the proof of Theorem 2 we shall construct a suitable total variation functional $\mathrm{ProbTV}_\Psi$ which upper-bounds $J_\Psi$.

The next theorem asserts existence of soft classifiers for the regularized risk minimization using $J_\Psi$ for very general functions $\Psi$ and hypothesis classes $\mathcal{H}$, requiring only that $\Psi$ be lower semicontinuous. For example, every continuous function and also $\Psi(t) = \mathbf{1}_{t>p}$ for $p \in [0,1]$ satisfies this. The existence theorem is valid for all hypotheses classes which are closed in a suitable sense.

**Theorem 4.** *Under Assumption 1, for every lower semicontinuous function* $\Psi : [0,1] \to [0,1]$*, and whenever* $\mathcal{H}$ *is a weak-\* closed hypothesis class of* $\mathcal{A}$*-measurable functions* $u : \mathcal{X} \to [0,1]$ *in the sense of Definition 1 in the appendix, there exists a solution to the problem*

$$\inf_{u \in \mathcal{H}} \mathbb{E}_{(x,y) \sim \mu} \left[|u(x) - y|\right] + J_\Psi(u).$$

**Example 1.** Let us consider three interesting hypothesis classes of weak-\* closed classifiers for which Theorem 4 applies. More detailed explanations are given in Appendix A.8.

1.  The simplest such class $\mathcal{H}$ is the class of *all* $\mathcal{A}$-measurable soft classifiers $u : \mathcal{X} \to [0,1]$ which could be referred to as *agnostic* classifiers since they are not parametrized.

2.  An example with more practical relevance is the class of (feedforward or residual) neural networks defined on the unit cube $\mathcal{X} := [-1,1]^d$ with uniformly bounded parameters

    $$\mathcal{H} := \Big\{ \Phi_L \circ \cdots \circ \Phi_1 : [-1,1]^d \to [0,1] : \Phi_l(\bullet) = A_l \bullet + \sigma_l(W_l \bullet + b_l),$$
    $$\|(A_l, W_l, b_l)\| \leq C \; \forall l \in \{1, \ldots, L\} \Big\},$$

    where we assume that the activations $\sigma_l : \mathbb{R} \to \mathbb{R}$ are continuous. Note that the boundedness of the weights cannot be relaxed. To see this, consider the (very simplistic) neural network $u_n(x) = \tanh(w_n x)$ for $x \in [-1,1]$ and $w_n \in \mathbb{R}$. For $w_n \to \infty$ it is easy to see that $u_n$ converges to $u(x) := \mathrm{sign}(x)$ which does not lie in the same hypothesis class.

3.  Finally, one can also consider the class of hard linear classifiers on $\mathbb{R}^d$. Letting $\theta(t) := \mathbf{1}_{t>0}$ denote the Heaviside function, this class is given by

    $$\mathcal{H} := \big\{ \theta(w \cdot x + b) \, : \, w \in \mathbb{R}^d, \; |w| = 1, \; b \in [-\infty, \infty] \big\},$$

    where one interprets $u(x) := \theta(w \cdot x + b)$ as $u \equiv 1$ if $b = \infty$ and $u \equiv 0$ if $b = -\infty$. If the distributions $\rho_0$, $\rho_1$, and $\mathfrak{p}_x$ are sufficiently nice, then $\mathcal{H}$ has the desired closedness property.

## 2.4 Properties and asymptotics of $\mathrm{ProbPer}_\Psi$

In this section we shall discuss the interpretation of the functional $\mathrm{ProbPer}_\Psi$ defined in (13) as a *perimeter*. We do this in two ways.

First, we focus on the case where $\Psi$ is concave and non-decreasing and prove that $\mathrm{ProbPer}_\Psi$ is a *submodular functional*. If, in addition, $\Psi$ is assumed to satisfy $\Psi(0) = 0$, then $\mathrm{ProbPer}_\Psi(\mathcal{X}) = \mathrm{ProbPer}_\Psi(\emptyset) = 0$. Following Chambolle et al. [2015], for $\Psi$ satisfying these properties one can interpret $\mathrm{ProbPer}_\Psi$ as a generalized perimeter, i.e., a functional that can be used to measure the "size" of the boundary of a set. In Appendix A.3 we introduce $\mathrm{ProbPer}_\Psi$'s induced (generalized) total variation and use it in the proof of Theorem 2; note that, as discussed by Bungert et al. [2023], the adversarial problem (5) also induces a generalized perimeter with associated total variation.

**Theorem 5.** *If $\Psi(0) = 0$, then $\mathrm{ProbPer}_\Psi(\mathcal{X}) = \mathrm{ProbPer}_\Psi(\emptyset) = 0$. If $\Psi$ is concave and non-decreasing, then the functional $\mathrm{ProbPer}_\Psi$ is submodular, meaning that*

$$\mathrm{ProbPer}_\Psi(A \cup B) + \mathrm{ProbPer}_\Psi(A \cap B) \leq \mathrm{ProbPer}_\Psi(A) + \mathrm{ProbPer}_\Psi(B) \quad \forall A, B \in \mathcal{A}.$$

**Example 2.** For $\Psi(t) = t$ our perimeter reduces to the perimeter on the *random walk space* $(\mathcal{X}, \mathfrak{p})$, introduced by Mazón et al. [2020]: $\mathrm{ProbPer}_\Psi(A) = \int_{\mathcal{X} \setminus A} \int_A \mathrm{d}\mathfrak{p}_x \, \mathrm{d}\rho_0(x) + \int_A \int_{\mathcal{X} \setminus A} \mathrm{d}\mathfrak{p}_x \, \mathrm{d}\rho_1(x)$.

Second, we consider more general $\Psi$ and show that $\mathrm{ProbPer}_\Psi$ is related to a standard *local* perimeter when the adversarial budget approaches zero; for the case of adversarial training such a connection was proved by Bungert and Stinson [2022] where the authors utilized the notion of Gamma-convergence of functionals. We take a first step in this direction by proving that for sufficiently smooth sets the probabilistic perimeter converges to a local one if the family of probability distributions $\mathfrak{p}_x$ localizes suitably. For example, one could think of $\mathfrak{p}_x := \mathrm{Unif}(B_\varepsilon(x))$, which converges to a point mass at $x$ if $\varepsilon \to 0$. To make our setting precise, we pose the following general assumption:

**Assumption 2.** We assume that $\mathcal{X} = \mathbb{R}^d$, $\Psi(0) = 0$, $\Psi$ is measurable and bounded, and $\rho_1, \rho_0$ have continuous densities with respect to the Lebesgue measure which we shall also denote as $\rho_1, \rho_0$. Furthermore, we assume that there is $\varepsilon > 0$ and a measurable function $K : \mathcal{X} \times \mathbb{R}^d \to [0, \infty)$ such that for every $x \in \mathbb{R}^d$ we have the representation

$$\mathrm{d}\mathfrak{p}_x(x') = \varepsilon^{-d} K\left(x, \frac{x' - x}{\varepsilon}\right) \mathrm{d}x'.$$

We also assume that for every $x \in \mathcal{X}$ we have $K(x, \bullet) \in L^1(\mathbb{R}^d)$, $\int_{\mathbb{R}^d} K(x, z) \, \mathrm{d}z = 1$, and $K(x, z) = 0$ if $|z| > 1$, and that for every $z \in \mathbb{R}^d$ the mapping $x \mapsto K(x, z)$ is $C^1$.

**Proposition 1.** *Under Assumption 2, if $A$ has a compact $C^{1,1}$ boundary and either $\Psi$ is continuous or there exists a constant $c > 0$ such that $K(x, z) \geq c$ for all $x \in \mathcal{X}$ and $|z| \leq 1$, then*

$$\lim_{\varepsilon \to 0} \frac{1}{\varepsilon} \mathrm{ProbPer}_\Psi(A) = \int_{\partial A} \sigma_{0,\Psi}[x, n(x)] \rho_0(x) + \sigma_{1,\Psi}[x, n(x)] \rho_1(x) \, \mathrm{d}\mathcal{H}^{d-1}(x) \quad (17)$$

*where we let $n(x)$ denote the normal to $\partial A$ at a point $x \in \partial A$, and for any vector $v \in \mathbb{R}^d$ we define*

$$\sigma_\Psi^0[x, v] := \int_0^1 \Psi\left(\int_{\{z \cdot v \leq -t\}} K(x, z) \, \mathrm{d}z\right) \mathrm{d}t, \quad \sigma_\Psi^1[x, v] := \int_0^1 \Psi\left(\int_{\{z \cdot v \geq t\}} K(x, z) \, \mathrm{d}z\right) \mathrm{d}t.$$

**Remark 1.** If $K$ is radially symmetric and independent of $x \in \mathcal{X}$, then $\sigma_\Psi^0 = \sigma_\Psi^1 =: \sigma_\Psi$ is just a constant. E.g., for $K(x, z) := |B_1(0)|^{-1} \mathbf{1}_{|z| \leq 1}$ and $\Psi(t) = \mathbf{1}_{t > p}$ it is trivial that for $p = 0$ we have $\sigma_\Psi = 1$. However, for $p \geq \frac{1}{2}$ one easily sees $\sigma_\Psi = 0$, hence the limiting perimeter equals zero and there is no regularization effect. Using the function $\Psi(t) = \min\{t/p, 1\}$ corrects this degeneracy.

Notably, for radially symmetric $K$ the limiting perimeter in (17) coincides, provided $\sigma_\Psi > 0$, with the one derived for adversarial training (problem (5)) by Bungert and Stinson [2022], although they considered more general (potentially discontinuous) densities $\rho_i$. In particular, our result indicates that for very small adversarial budgets the regularization effect of both probabilistically robust learning and adversarial training is dominated by the perimeter in (17). While Proposition 1 already completes half of the proof (namely the limsup inequality) of Gamma-convergence of $\frac{1}{\varepsilon} \mathrm{ProbPer}_\Psi$ to the limiting perimeter, the remaining liminf inequality is beyond the scope of this paper. Proving that the convergence (17) does not only hold for sufficiently smooth sets as assumed in Proposition 1 but even in the sense of Gamma-convergence is an extremely important topic for future work since only Gamma-convergence allows to deduce from the convergence of the perimeters that also the solutions of probabilistically robust learning converge to certain regular Bayes classifiers as $\varepsilon \to 0$, see Bungert and Stinson [2022, Section 4.2].

## 3 General models

We now shift our attention to training general hypotheses $h \in \mathcal{H}$ using general loss functions $\ell : \mathcal{Y} \times \mathcal{Y} \to \mathbb{R}$. Motivated by Theorems 1 and 3 we propose the following probabilistically robust optimization problem:

$$\inf_{h \in \mathcal{H}} \mathbb{E}_{(x,y) \sim \mu} \left[\max\left\{p\text{-}\operatorname*{ess\,sup}_{x' \sim \mathfrak{p}_x} \ell(h(x'), y), \ell(h(x), y)\right\}\right]. \quad (18)$$

In the mathematical finance or economics literature the $p$-ess sup operator is better known as the value at risk (VaR) of a random variable at level $p$ and it is notoriously hard to optimize. VaR is closely related to other risk measures like, for instance, the conditional value at risk (CVaR) which is convex and easier to optimize [Robey et al., 2022, Rockafellar et al., 2000]. For a function $f : \mathcal{X} \to \mathbb{R}$ and a probability distribution $\mathfrak{p}$ the CVaR at level $p$ is defined as

$$\mathrm{CVaR}_p(f; \mathfrak{p}) := \inf_{\alpha \in \mathbb{R}} \alpha + \frac{\mathbb{E}_{x' \sim \mathfrak{p}_x} \left[ (f(x') - \alpha)_+ \right]}{p}. \tag{19}$$

It is easy to see that $p\text{-}\operatorname{ess\,sup}_{x' \sim \mathfrak{p}} f(x') \leq \mathrm{CVaR}_p(f; \mathfrak{p})$. Using CVaR in place of the $p\text{-}\operatorname{ess\,sup}$ operator, a tractable version of (18) is

$$\inf_{h \in \mathcal{H}} \mathbb{E}_{(x,y) \sim \mu} \left[ \max \left\{ \mathrm{CVaR}_p(\ell(h(\bullet), y); \mathfrak{p}_x), \ell(h(x), y) \right\} \right]. \tag{20}$$

We emphasize that, if the loss function $\ell(\bullet, \bullet)$ is convex in its first argument, then (20) is a convex function of the hypothesis $h$. Furthermore, CVaR is positively homogeneous and hence also (20) is positively homogeneous in the loss function. So, taking the maximum of the samplewise CVaR and standard risk is meaningful as both terms scale in the same way.

In the binary classification case we can prove the following interesting result that the CVaR relaxation corresponds precisely to using the risk $\mathrm{ProbR}_\Psi$ with a special piecewise linear and concave function $\Psi$ for which our theory from Section 2.2 applies. In Appendix A.5 we prove a more general version of the following statement, replacing the $[\,\bullet\,]_+$ operation in (19) with a `Leaky ReLU`.

**Theorem 6.** *Let the function* $\Psi_p : [0, 1] \to [0, 1]$ *be defined as* $\Psi_p(t) := \min\{t/p, 1\}$. *Then it holds*

$$\mathrm{CVaR}_p \left( \mathbf{1}_{\mathbf{1}_A(\bullet) \neq y}; \mathfrak{p} \right) = \Psi_p \left( \mathbb{P}_{x' \sim \mathfrak{p}} \left[ \mathbf{1}_A(x') \neq y \right] \right)$$

*and as a consequence for all* $A \in \mathcal{A}$:

$$\mathbb{E}_{(x,y) \sim \mu} \left[ \max \left\{ \mathrm{CVaR}_p(\mathbf{1}_{\mathbf{1}_A(\bullet) \neq y}; \mathfrak{p}_x), \mathbf{1}_{\mathbf{1}_A(x) \neq y} \right\} \right] = \mathrm{ProbR}_{\Psi_p}(A).$$

An immediate consequence of Theorem 6 is that for binary classification (20) has a solution.

**Corollary 1.** *Under Assumption 1 and in the setting of Theorem 6 problem* (20) *has a solution.*

In Appendix A.5 we collect a few more observations concerning the CVaR, especially focussing on its behavior for $p > 1$. These geometric properties, the homogeneity with respect to the loss function, its potentially favorable sample complexity (see the discussion in Appendix A.6), and its versatility for algorithmic implementation make (20) a notable generalization of the adversarial training problem (2). Notice that when $p \to 0$ one formally recovers (2) from (20).

# 4 Numerical results

We build upon the code of Robey et al. [2022]. The algorithmic realization of (20) is a straightforward adaptation of their algorithm, which alternatingly minimizes the inner optimization problem that defines CVaR and the outer optimization to find a suitable classifier, see Algorithm 1 in Appendix B. In our experiments, we conduct a comparative analysis between their algorithm (denoted as "Original" in Table 1) and Algorithm 1 in the appendix which is based on (20) (denoted as "Geometric"). We report the clean, and adversarial accuracies (subject to PGD attacks), as well as accuracies on noise-augmented data and quantile accuracies for different values of $p$ (see [Robey et al., 2022, (6.1)] for the definition) averaged over three runs; see Appendix B.2 for more training details. Our experiments are conducted on MNIST and CIFAR-10 and to ensure a fair comparison we adhere to the hyperparameter settings described by Robey et al. [2022], such that both the original and geometric algorithms utilize the same set of hyperparameters for each specified value of $p$. The corresponding results for several baseline algorithms including empirical risk minimization and adversarial training can be found in their paper. We perform model selection based on the best clean validation accuracy. The results in Table 1 show that for moderate values of $p$ our geometric modification induces higher adversarial robustness than the original PRL without loss of clean accuracy (see, in particular, the results for MNIST with $p = 0.1$ and for CIFAR-10 with $p = 0.3$). In the noise augmented metrics as well as for extreme values of $p$ close to 0 or equal to 0.5 both algorithms behave comparably. The latter can be expected from out theoretical results, in particular Proposition 1.

Note that the original or the geometric version of PRL should not be expected to match the adversarial robustness of classifiers trained with PGD attacks [Madry et al., 2017] or other worst-case optimization techniques. Instead, they shine with superior clean accuracies and easier training while maintaining probabilistic and a certain degree of adversarial robustness, as also observed by Robey et al. [2022].

We also remark that our sweep over different values of $p$ confirms that increasing this parameter interpolates between low and high clean accuracies. However, it should be noted that it does not necessarily result in a direct interpolation between high and low adversarial or probabilistic accuracy, as claimed by Robey et al. [2022]. These observations hold true for both the original algorithm and our geometric modification, and despite utilizing their code and hyperparameters, we were unable to reproduce the exact results reported by Robey et al. [2022, Tables 1-4].

Table 1: Accuracies [%] of the geometric and original algorithm for different values of $p$.

| Data | $p$ | Algorithm | Clean | Adv | Aug | Aug-0.1 | Aug-0.05 | Aug-0.01 |
|------|-----|-----------|-------|-----|-----|---------|----------|----------|
| MNIST | 0.01 | Geometric | **99.20** | **12.19** | 99.04 | 98.18 | 97.69 | 96.38 |
| | | Original | 99.19 | 10.76 | 98.90 | 97.94 | 97.38 | 95.67 |
| | 0.1 | Geometric | 99.28 | **14.20** | 99.22 | 98.70 | 98.45 | 97.86 |
| | | Original | **99.32** | 8.94 | 99.22 | 98.70 | 98.46 | 97.80 |
| | 0.3 | Geometric | **99.29** | **3.02** | 99.21 | 98.76 | 98.53 | 97.95 |
| | | Original | 99.27 | **3.02** | 99.22 | 98.77 | 98.55 | 98.01 |
| | 0.5 | Geometric | **99.27** | **1.80** | 99.21 | 98.72 | 98.44 | 97.93 |
| | | Original | 99.26 | 1.68 | 99.19 | 98.72 | 98.47 | 97.80 |
| CIFAR-10 | 0.01 | Geometric | 80.65 | 0.15 | 78.13 | 73.44 | 72.13 | 68.80 |
| | | Original | **81.73** | **0.24** | 79.16 | 74.61 | 73.19 | 69.96 |
| | 0.1 | Geometric | 88.15 | 0.14 | 85.96 | 82.55 | 81.46 | 78.81 |
| | | Original | **88.28** | **0.19** | 85.61 | 82.21 | 81.06 | 78.28 |
| | 0.3 | Geometric | **90.43** | **11.80** | 88.70 | 85.17 | 83.93 | 80.93 |
| | | Original | 89.97 | 7.20 | 88.62 | 85.07 | 83.75 | 80.87 |
| | 0.5 | Geometric | **91.51** | 1.93 | 88.94 | 85.53 | 84.18 | 81.21 |
| | | Original | 90.74 | **1.99** | 88.94 | 85.54 | 84.35 | 81.57 |

## 5  Discussion and Conclusion

In this paper we considered probabilistically robust learning (PRL), originally proposed by Robey et al. [2022]. We corrected a subtle but crucial theoretical flaw in the original formulation by introducing a regularization of the standard risk with nonlocal perimeters measuring the susceptibility of the decision boundary towards high-probability adversarial attacks. For binary classification we proved existence of optimal hard classifiers and of very general classes of soft classifiers including neural networks. We also provided an asymptotic expansion for smooth decision boundaries to show that for small adversarial budgets the probabilistic perimeters discussed in the paper induce the same regularization effect as adversarial training. For general (not necessarily binary) problems we showed that the natural loss function to choose is the sample-wise maximum of the standard loss and conditional value at risk (CVaR).

One limitation of PRL is that it does not completely solve the accuracy vs. robustness trade-off, which remains a challenging problem. Furthermore, while the formal limit of PRL as $p \to 0$ is the worst-case adversarial problem, the algorithms for solving PRL exhibit limitations for very small values of $p$ (in the computation of $\text{CVaR}_p$). Still, the results for moderately large values of $p$ are encouraging and future work should focus on understanding of this trade-off better.

The rich mathematical theory developed in this paper opens up new avenues for research, such as the explicit design of probabilistic regularizers for algorithms and exploring the variational convergence of the probabilistic perimeter and its implications for adversarial robustness.

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
