# OpenReview forum: "It begins with a boundary: A geometric view on probabilistically robust learning"
_NeurIPS.cc/2023/Conference — Submitted to NeurIPS 2023_

### Official Review · Reviewer_CsNz · 2023-06-09

**Soundness:** 3 good
**Presentation:** 3 good
**Contribution:** 2 fair
**Rating:** 3
**Confidence:** 5

**Summary:**

This paper claims to identify a flaw in the formulation of probabilistically robust learning.  To correct this supposed flaw, the authors propose a regularization scheme based on a notion of probabilistic perimeter.  This leads to a set of theoretical results that generalize the losses used in probabilistic robustness, identify conditions under which solutions exist, and initiate a theory of the limiting cases of these generalizations.  Experiments on MNIST and CIFAR-10 are also provided, showing that algorithmically, this new approach performs almost exactly the same as the previously proposed CVaR approach.

**Strengths:**

**New generalizations of probabilistic robustness.**  This paper rigorously analyzes the framework of probabilistic robustness, and provides several avenues for future research.  The perimeter-based perspective gives a new interpretation; it feels almost, but not necessarily the same as saying that this paper is to the original PRL paper as TRADES is to vanilla adversarial training (e.g., PGD).  I think that mathematically, this paper presents a novel and deep set of results which may be of interest to others in the community.  Furthermore, the idea of introducing the probabilistic perimeter is relatively clean, in the sense that it seems natural to consider the ProbR(A) function as a decomposition of the probabilistic risk and the clean risk.

**Existence of solutions.**  The existence proofs are interesting and general here.  In particular, I found the insight that the CVaR problem can be directly related to a generalized perimeter problem in Theorem 6 to be surprising.  Although it was previously known that solutions existed in special cases, this result is much more general.

**Writing.**  In general, I found the writing in this paper to be relatively strong.  I would tend to soften some of the language, such as referring to various results as "astonishing" and calling various results "extremely important," but at the end of the day, these are stylistic choices that can be made at the discretion of the authors.

**Weaknesses:**

### Contributions

I'll start by evaluating the extent to which this paper lives up to the contributions it claims to make in Section 1.3.  In particular, I am going to focus on the first and last contributions; the middle two contributions are solid in my opinion.

**Contribution 1.**  The first claimed contribution is as follows:

> "We address the geometric limitation of the model by Robey et al. [2022] by introducing a family of perimeter regularizations."

Throughout the paper, the authors make several related claims regarding the validity of the approach outlined in [Robey et al. 2022], e.g.,

> "We propose a geometric framework for understanding PRL, which allows us to identify a subtle flaw in its original formulation and to introduce a family of probabilistic nonlocal perimeter functionals to address this."

In order to validate this contribution, it is essential that the authors

1. Clearly articulate which property PRL is missing;
2. Explain why missing this property constitute a flaw; and
3. Show that their perimeter-based method addresses this flaw.

In this respect, I'm not sure whether any of these three points are satisfied.  I'll go through each of these points, and then hopefully we can have a discussion about this during the rebuttal phase.  To this end, let's start by looking into the claims of this PRL paper.  Based on my reading, the goal of that work is to do the following (as quoted from the introduction of [Robey et al., 2022]):

> "The fundamental drawbacks of these learning paradigms motivate the need for a new robust learning framework that (i) avoids the conservatism of adversarial robustness without incurring the brittleness of ERM, (ii) provides an interpretable way to balance nominal performance and robustness, and (iii) admits an efficient and effective algorithm. To this end, in this paper we propose a framework called probabilistic robustness that bridges the gap between the accurate, yet brittle average-case approach of ERM and the robust, yet conservative worst-case approach of adversarial training. By enforcing robustness to most rather than to all perturbations, we show theoretically and empirically that probabilistic robustness meets the desiderata in (i)–(iii)"

In short, the goal seems to be to design a learning framework that enforces robustness to most of the perturbations in a ball around each data point.  Now, returning to the paper under review, the supposed flaw is described as follows:

> "We highlight that [the PRL model] does not constitute a non-negative functional of A. Hence the loss function in (6) is not a regularized version of the standard risk (4) and in fact can be strictly smaller. This observation reveals a subtle flaw in the approach of Robey et al. [2022]: Points which lie in thin or spike-like regions of A penetrating the other class and that are more likely to have the label zero than the label one (meaning they lie in the set $[\rho_0 > \rho_1]$) yield negative contributions in (7) and are hence favored.  Such a scenario is visualized on the left side of Figure 1. From an adversarial perspective this means that points which are already misclassified are attacked nevertheless, which can lead to the bizarre situation that the adversary helps the learner by putting these points in the correct class with high probability, thereby reducing both adversarial robustness and clean accuracy."

Below I'll list some of my thoughts about this, but first, it seems important to agree on what constitutes a "flaw."  It seems reasonable to define a flaw to be **an inherent limitation or weakness that affects the validity, reliability, or applicability** of a particular method.  Hopefully the authors agree with this definition, because I'm going to base my discussion of this on this definition (and if not, we can hash it out in the rebuttal).  Here are my thoughts vis-a-vis this definition.

* *Non-negative functional.* Unless I missed something, it's not clear to me whether [Robey et al., 2022] claimed that their framework constituted a "non-negative functional of A."  Or, zooming out, they did not claim that it was essential to their paradigm to ensure that the clean data was classified correctly.  While they do measure the clean accuracy in their experiments, their paradigm does not seem to be designed to explicitly minimize the standard risk or any regularized form of the standard risk.  On the contrary, the aim seems to be to enforce robustness to "most" of the perturbations in a ball around each data point, which more or less seems to be accomplished in the PRL paper.  Therefore, the motivation based on the non-negative functional perspective underlying the supposed flaw seems to be in doubt.  Or in other words, can a feature of paradigm constitute a flaw if that paradigm never claimed that feature as a contribution?

* *Spike-like regions.* The next issue is that the authors do not define what they mean by "spike-like" regions.  This confusion is amplified by Figure 1.  In this figure, why is PRL being applied to a non-robust classifier?  And what do the green and yellow regions correspond to?  Are the regions $\epsilon$-expansions of the decision boundary?  If this is the case, then why are these regions not a uniform distance away from the decision boundary?  And why are some points in red on the wrong side of the decision boundary?  Without *clearly* identifying the problem, it's difficult for the reader to accept that a flaw has truly been found.

* *Label probabilities.* I don't understand how the idea that the data points which are "most likely to have the label zero than the label one" yield negative contributions to the standard risk.  Can the authors explain this further, or can you illustrate this set $\{x : \rho_1(x) > \rho_0(x)$ in Figure 1?  And then how is this impacted by the perturbations added to the instance? The picture that I (perhaps incorrectly) have in mind is Figure 1 from the randomized smoothing paper by Cohen et al. (https://arxiv.org/pdf/1902.02918.pdf), although I'm having trouble mapping back onto the proposed approach.

* *Adversarial perspective.* The next point worth discussing is the "adversarial perspective" mentioned above.  Based on my understanding of PRL, there is no adversary.  The learner simply receives a distribution over perturbations and attempts to be robust against most of the perturbations sampled from that distribution.  And so from this perspective, it's unclear how the adversary factors in.  Thus, I don't understand the so-called "bizarre situation" that the authors claim to identify, or why it would serve to reduce both clean and adversarial accuracy.  Could the authors elaborate here?

To summarize, it's unclear to me what the supposed flaw is and why this flaw constitutes an inherent limitation of the method.  And given this, the contribution involving introducing a new family of "perimeter regularization" is unclear, because it's not clear what flaw or problem this solution is addressing.  More generally, I think it's worthy being wary of claiming to have found a flaw in existing work unless that flaw can be clearly identified.  While scientific research should certainly scrutinize past work, it seems important to charitably acknowledge what that past work claims -- and dually, what it does not claim -- to accomplish.  And in this case, my understanding of the argument laid out by the authors is that it is somewhat parallel to the ideas laid out in PRL, in that the supposed flaw is not something that the PRL framework was ever intended to do.  However, it could be the case that I have misunderstood either the author's argument or the original PRL paper, and therefore a discussion of this would be welcome.

**Contribution 4.**  The final contribution is as follows:

> "We extend our models to encompass general loss functions and hypothesis classes. Our numerical experiments demonstrate that our geometric correction can enhance the adversarial robustness of probabilistically robust classifiers without compromising clean accuracy."

I believe that this is inaccurate.  Based on Table 1, it seems that around half of the time, the original PRL method actually achieves better adversarial robustness than the proposed geometric method.  Moreover, the clean accuracy of models trained with the original PRL method seem to be either better or within a single percentage point of the accuracies reported for the geometric variant.  Therefore, it seems at best inconclusive as to whether this approach actually results in better performance.

---

### **Notation**

**Nested subscripts.**  The notation in this paper is at times difficult to parse.  I find it particularly hard to parse expressions like (9) and (10).  I think that it would be easier to read if the authors used expectation notation, and avoided the up and down carets for and/or.  For example, the first term in (10) could be written as

$$
\mathbb{E} [ x\in A \quad\text{and}\quad \text{Pr}_{x'\sim \mathbb{P}_x}[x'\in A] > p ]
$$

and one could perhaps add $x\sim\rho_0$ as an underscore (LaTeX in openreview isn't rendering this underscore correctly).  This would be much clearer, and it would prevent the use of three levels of underscores, e.g., $1_{\mathbb{P}_{x'\sim p_x}}$ in (10).

**Definition of the probabilistic perimeter.**  I think that the definition of the probabilistic perimeter could be clearer.  Overall, I'm not sure what is gained by reducing the problem to the setting of binary classification and considering sets $A$.  (8) defines ProbPer in this special case, but to me, it seems clearer to define it as follows:

$$
\text{ProbPer}(f) = \text{Pr}_{x,y} [  f(x)\neq y \quad\text{and}\quad \text{Pr}[f(x+\delta) = y] > p ]
$$

where $f$ is a classifier and the inner probability is taken WRT $\mathfrak{p}_x$.  This avoids the somewhat confusing $\rho_0$ notation and naturally extends to multi-class classification settings.

### **Related work**

**Other interpolation methods.**  It would be worth citing other recent relevant work on robust learning between the average and worst case.  Two notable papers, which are compared to in the PRL paper, are

> Rice, Leslie, et al. "Robustness between the worst and average case." Advances in Neural Information Processing Systems 34 (2021): 27840-27851.
> Li, Tian, et al. "Tilted empirical risk minimization." arXiv preprint arXiv:2007.01162 (2020).

One could also think of the TRADES paper as interpolating between adversarial training and ERM through the trade-off parameter $\beta$, so it would be worth comparing to this paper as well.

**Engagement with PRL.**  As discussed above, it would be worth softening the claims regarding the extent to which PRL has fundamental flaws.  In addition, the authors make several other claims in the same spirit which don't necessarily agree with past work.  For instance,

> "To the best of our knowledge, existence of solutions for (6) or even (3) has not been proved so far."

Having read the PRL paper, this seems to be untrue.  In [Robey et al., 2022], the authors derive closed form expressions for PRL in linear regression and mixture-of-Gaussians classifications.  This latter scenario is particularly relevant given that the authors restrict their attention for much of the paper to binary classification.  Therefore, I think it may make sense for the authors to revise this claim.

### **Experiments**

**Introducing upper bounds on the 0-1 loss.** I'm not sure I understand the value in introducing the upper bound $\Psi$ in the paragraph starting on line 145.  The authors do not conduct any experiments (as far as I can tell) using this formulation.  So the question remains: Is there any practical/empirical value in using such a surrogate?

### **Theory**

**What constitutes a theorem?**  It's worth broaching what can at times be a sensitive topic: What constitutes a theorem?  I would argue that Theorem 1 and 3 are closer to being remarks than they are to being theorems, although I could see counterarguments.  The fact is that these results hold essentially by construction, whereas as results such as Theorem 2 are quite non-trivial.

**What is weak-*?**  It would be helpful if the authors could offer some of the preliminaries necessary to understand their results.  For instance, I do not have really any understanding of what weak-* means (even after reading definition 1) and how it differences form other notions of convergence (e.g., in the L^p sense). Moreover, I expect that most of the people reading this paper will be in the same boat.  This not an issue per se, but it's worth considering that you may lose a good chunk of your audience by Section 2.3 due to the inaccessibility of some of this math.  One way to ameliorate this would be to offer an intuitive explanation of what weak-* convergence means and to point to references where one could learn about what weak-* means, or to define weak-* in the appendix somewhere.  If it only appeared in the appendix, I would probably not have noticed it, but since it features relatively prominently in Theorem 4 and Example 1, it's definitely worth discussing and adding some intuition before jumping into examples.  Another way of helping the reader to understand the classes of hypotheses that mean the weak-* closed condition would be to give an example or two that do *not* satisfy this property.

**Why are the properties and asymptotics of PropPer interesting?**  The contribution of the Section 2.4 is unclear to me.  The fact that under certain conditions, $\text{ProbPer}_\Psi$ is submodular doesn't seem to have a punchline, i.e., it's unclear why this property is important and/or why one would want this result to hold.  Perhaps the authors could elaborate here on why submodularity is desirable, or what properties it engenders on the task of optimizing ProbPer?

A similar criticism applies to Proposition 1.  The authors say that this result speaks to the fact that for adversarial training and probabilistic robustness, the perimeter dominates the regularization.  Again, I'm left wondering, why is this notable?  Does this lead to a better understanding of how to optimize ProbR(A)?  I can see why one might want it to be the case that PRL converges to the Bayes classifier as $\epsilon\to zero$, but again, it's not clear to me why such a result would be "extremely important."  One way to make this more clear would be to elaborate more on the implications listed in the paragraph starting on line 269.

### **Miscellaneous**

Here are a few other points:

* I think that "Machine Learning" doesn't need to be capitalized on page 1, nor does "Probabilistically Robust Learning."
* There is a heavy reliance of $p$ and similar looking characters here.  We have $p$ as the risk probability, $\mathfrak{p}_x$ as the distribution over perturbed instances, $\mathbb{P}$ as the probability function, and $\rho$ as a measure over a particular class.  It would improve readability to use characters that look more different from one another.
* What is meant by an *admissible* set in line 126?  It doesn't seem relevant to statistical notions of admissibility unless I'm missing something.
* The comma after neighbors on line 127 should be removed.
* On line 174, it should be *an* $\mathcal{A}$-measurable function.

### **Overall evaluation**

Overall, I thought that there were some interesting insights in this paper.  The formulation is relatively clean and there are several interesting new insights about probabilistic robustness.  The writing is generally of high quality and some of the new results are surprising and generalize past results.

However, there are also numerous drawbacks.  I can't help but feel that this paper -- as it is currently written -- will have a limited impact.  Many may read it and come to the conclusion that this paper does a lot of interesting math and generalizes notions of probabilistic robustness.  But at the same time, these readers may wonder: Why did we do all of this?  Or, in other words, do these generalizations offer any insights into how one could achieve better probabilistic robustness in practice.  And unfortunately, I don't see how one could answer this question in the affirmative.  The experiments do not demonstrate the benefit of the vanilla geometric or the generalized variants (e.g., soft classifiers, smoothed losses $\Psi$, etc.) of probabilistic robustness.  This is all to say that I think that this would be a much stronger paper if there was a resounding demonstration that it's actually worth generalizing probabilistic robustness, and/or whether or not the supposed flaw actually holds back the original framework.

This is related to my feeling that there isn't an inherent flaw in the original framework.  I was not convinced by the arguments given by the authors (as explained above) and I think that it would be worth rethinking the claims to having identified this flaw, especially given that the authors are in a sense focusing on a feature that was never claimed to be a component of probabilistic robustness (based on my limited understanding).  This doesn't necessarily dilute the contribution; it would simply serve to engage with the past work in a different way.

Overall, I think that this paper is borderline, and as written I would tend to recommend that it not be accepted on the basis that all of the mathematical insights given in the main text do not lead to any notable improvements on the algorithmic end.  If the authors can demonstrate that these insights do lead to better algorithms, and if we can work through some of the topics discussed above, I would consider raising my score.

**Questions:**

See above.

**Limitations:**

I don't see any major limitations.

---

> ### Author Rebuttal · Authors · 2023-08-09
>
> We thank the reviewer for their detailed comments! In our response we follow the structure of the review and give point to point replies.
>
> ----
> **Overall evaluation**
> - *The mathematical insights $\ldots$ don't lead to $\ldots $ notable improvements on the algorithmic end.*
>  We note that the goal of mathematics (even in the context of ML) is not solely to improve algorithms; uncovering structure and drawing connections to new areas are equally essential. Our analysis mainly serves to understand, improve, and justify the modeling behind PRL. The goal of our experiments is to show that our modifications (which, as we argue, lead to stronger/cleaner/more interpretable theoretical guarantees) **do not decrease empirical performance**. We think this is not a minor observation, as usually there is a trade-off between strong theoretical guarantees and empirical performance. We also note that, as stated on the NeurIPS call for papers, works that advance either theoretical or algorithmic aspects of ML are welcomed.
>
> ----
> **Contributions**
> - *Non-negative functional.*  You're right, this is not claimed by Robey et al; but nowhere in our paper do we suggest this. This geometric interpretation is a key part of our contribution: viewing the loss as a regularization of standard risk is a transparent and interpretable way of showing that the pathological decision boundaries highlighted in Fig. 1 (see also attached figure) cannot minimize our proposed loss.
> - *Robey et al did not claim $\ldots$ was classified correctly.* To clarify, we show that the PRL model intriguingly allows for attacks which "correct" misclassified **training data**. However, this will strictly decrease accuracy on **test data** drawn from the same distribution, without necessarily improving robustness. Our simple modification eliminates this possibility. See global rebuttal.
> - *Spike-like regions.* An informal definition of spike-like regions is high-curvature regions of the classification boundary. In the context of PRL with a fixed budget $\varepsilon>0$ these are points $x\in A$ where most of the ball $B_\varepsilon(x)$ intersects the complement $A^c$. See the new figure for an illustration.
> - *Label probabilities: I don't understand how $\ldots$ yield negative contributions to the standard risk.* This issue is best explained in the new figure. A point with the label zero (blue) but lying in the thin region of the classifier $A$ does not increase the original PRL loss, which is zero. In contrast, the standard risk is strictly positive as the point is misclassified.
> - *Adversarial perspective.* While there is no explicit adversary in the PRL framework (in the sense that it's no minimax problem), we recall that in Robey et al it is repeatedly stated that PRL interpolates between robust and standard loss. Our small modification allows for an adversarial interpretation where the adversary samples a point from $\mathfrak p_x$ and either accepts it (if it yields a larger loss) or rejects it and sticks to the point $x$ otherwise. In contrast, the original PRL formulation is not adversarial since it always accepts the sample, irrespective of how it changes the loss.
>
> ----
> **Related work**
> - *Strength of language. Use of the word "flaw".* We see your point and will adjust our language. Instead of describing the difference between the original PRL and our model as a flaw, in the new version of the paper we will just describe properties that the two models do (not) share. PRL is a nice model and our goal is to build upon it and construct a geometric theoretical framework for its analysis.
> - *To the best of our knowledge, the existence of solutions $\ldots$ has not been proved so far. $\ldots$ This seems to be untrue, indeed Robey et al $\ldots$* We note that Robey et al only consider settings where the decision boundary is a hyperplane. This is restrictive since NNs often have complicated decision boundaries. As motivated by our discussion of thin spikes, interesting behavior arises when the boundary has curvature, see also the new figure. Our existence results Theorems 2 and 4 cover much more general cases (as acknowledged by the reviewer) like classifiers induced by neural networks. That said, we shall modify the language to better acknowledge Robey et al's contribution.
> - *One could also think of the TRADES paper as interpolating $\ldots$* Thanks for pointing us to this paper. We will discuss it in the new version.
>
> ---
> **Experiments**
> - *Is there any practical/empirical value in using such a surrogate?* Indeed this surrogate has high practical relevance since, as shown in Theorem 6, it's equivalent to using the conditional value at risk (CVaR) which is what Robey et al suggest to optimize instead of the $p$-$\operatorname{ess sup}$ operator.
>
> ---
> **Theory**
> - *What is weak-*?* We realize that the weak-* topology is not well-known outside the math community which is why we postpone all technicalities related to it to the appendix and only provide *relevant examples* for machine learning applications in the main text. An example for classifiers which *are not* weak-* closed is already contained in Example 1. The introduction of the weak-* topology is unavoidable in the present setting as it guarantees existence of minimizers if the space of admissible classifiers is *infinite dimensional*. We also notice that our proofs strongly use the *structure* of the modified PRL objective in an elegant and clean way (as other reviewers have highlighted).
> - *Why are the properties and asymptotics of PropPer interesting?* The properties of ProbPer are interesting for two reasons: The submodularity property is the analogue of *convexity* for set functionals and can be exploited to have better theoretical guarantees for optimization. The asymptotics are interesting because they show that both adversarial training and PRL as $\varepsilon\to 0$  recover the Bayes classifier which has the shortest decision boundary among all Bayes classifiers.

---

> > ### Comment · Reviewer_CsNz · 2023-08-21
> > **Rebuttal response**
> >
> > > "We note that the goal of mathematics. . . either theoretical or algorithmic aspects of ML are welcomed."
> >
> > While I appreciate the spirit of this, I feel strongly that theoretical insights without meaningful implications for the practice of ML constitutes a weaker contribution than one that advances both practical and theoretical aspects.  This may seem like a high bar, but given that the authors are proposing a new paradigm here, it's worth asking what the use of theory is when there isn't a clear reason to favor this paradigm over existing alternatives.  That is, the theory proposed here does not concern the existing practice of robustness; it pertains to a new approach, which is different from TERM, PRL, or TRADES, and therefore the theory should convince the reader that it's worth using this paradigm over existing alternatives.  And as outlined in the "Theory" section of my review, it's not clear (a) what the take-aways from these results are (in particular, those concerning submodularity, PropPer, etc.), and (b) how they will lead to more robust predictors.
> >
> > ---
> >
> > The biggest concern I have -- and it's a concern that I feel has not been addressed in the rebuttal -- is a straightforward answer to a question posed in my original review:
> >
> > > "To summarize, it's unclear to me what the supposed flaw is and why this flaw constitutes an inherent limitation of the method."
> >
> > The implication from the main argument of this paper is that elements such as the non-negative functional perspective or the "spike-like" regions contribute to PRL being flawed; however, these disparate elements do not yield a crisp identification for what this supposed flaw is.  And as the argument in this paper centers on correcting the supposed "flaw," my view on this matter has not changed since the original reviews.  Notably, it seems that the discussion has led to some consensus on this point; see, e.g., `Reviewer u3Qk`'s [response](https://openreview.net/forum?id=IvEWhB1P90&noteId=kmPtxnNZiH):
> >
> > > "Additionally, I support reviewer CsNz's request to engaging slightly differently with the previous work, i.e. to consider your own work as an extension/parallel development to address misclassified points rather than a specific flaw of the PRL framework."
> >
> > And while I think that Figure 1-2 in the rebuttal PDF are clearer than Figure 1 in the paper, it's still not clear to me how this indicates that the perimeter-based method is a better paradigm.  As the authors seem to acknowledge, the goal of PRL is to improve robustness; it is not designed to improve clear accuracy, although the authors of PRL seem to claim this as a side-effect.  So I think adjusting the language will make a difference here, although unfortunately it's hard to evaluate this given the organizers made the decision to not allow new drafts of the paper to be uploaded.
> >
> > WRT the adversarial perspective, the following sentence is still unclear to me:
> >
> > > "From an adversarial perspective this means that points which are already misclassified are attacked nevertheless, which can lead to the bizarre situation that the adversary helps the learner by putting these points in the correct class with high probability, thereby reducing both adversarial robustness and clean accuracy."
> >
> > The rebuttal doesn't clear up the conclusion I pointed to in my original review.  There is no "adversary to help the learner" in PRL (for $p\neq 0$) (c.f., "the original PRL formulation is not adversarial").
> >
> > ---
> >
> > Re: related work.  WRT the existence of solutions, the goal posts should not be changed.  The authors claimed that
> >
> > > "To the best of our knowledge, existence of solutions for (6) or even (3) has not been proved so far."
> >
> > The response, which argues that PRL considers hyperplane decision boundaries rather than DNN boundaries, is orthogonal to the claim made above.  The claim should either be made more specific -- i.e., it should specify that existence of solutions does not exist **for DNNs** -- or else reworded so that it acknowledges past contribution.

---

> > > ### Author Response · Authors · 2023-08-22
> > >
> > > What is discussed in our global rebuttal and in its attached illustration does constitute a crisp reason why one should consider introducing a modification to the original PRL model: the original PRL allows for minimizers that are neither robust nor accurate, even in the simplest scenarios. Furthermore, without our modification, one of the main arguments in favor of the PRL framework in Robey et al., namely, that it interpolates between standard risk minimization and adversarial training, is simply not correct; see Proposition 4 in our paper and its proof.
> > >
> > > We reiterate that our paper does not stop at proposing a modification of the original PRL model but it goes far beyond that. Indeed, we prove several theoretical results (existence of solutions in full generality, structural properties of the new objective functions, small budget asymptotics and connections to adversarial training, etc) that would not hold for the original PRL. We bring in this way multiple new insights onto PRL itself while introducing many interesting and novel techniques that may be used for the theoretical analysis of similar problems.
> > >
> > > Finally, we want to clarify that, contrary to what Reviewer CsNz states above, all other reviewers agree that the situations described in our global rebuttal indeed constitute an undesirable behavior of PRL. This is also the perspective of Reviewer u3Qk, who agrees with this point, as can be seen from their full response. Reviewer u3Qk centers their discussion around the likelihood of these bad instances arising in applications. As we have responded to them, we strongly believe that it is always better to propose principled models when possible. This is certainly one situation where this can be done easily, without additional computational effort and backed by theory.

---

### Official Review · Reviewer_LMsa · 2023-06-26

**Soundness:** 4 excellent
**Presentation:** 3 good
**Contribution:** 2 fair
**Rating:** 7
**Confidence:** 1

**Summary:**

Congratulations to authors for a timely and interesting paper. The paper advances the work of Robey et al. [2022] that established computational less expensive version of robust learning, dubbed Probabilistic Robust Learning (PRT). In particular, random perturbations and gradient based attacks had been used before and aside, while the work of [Robey 2022] proposed entirely probabilistic RT, which enjoys relatively lower computational costs.

The paper casts PRT as regularised Empirical Risk Minimisation. It further 1.) identifies a regularizer - 2.) improves it by introducing new objective $ProbR$ and 3.) relax it convex $\Psi$ instead of indicator function used in ProbR Further, the Section 2.3. 5 4.) Extends it to a soft classifiers (relevant to soft-max classification used in NNs).
Section 3 deals with general loss functions and upper bounds probabilistically robust objective by $CVaR$ (conditional Value at Risk) that is convex if loss is convex in the first argument (e.g. most of common losses used in NNs) and easier to optimise. Theorem 6 then establishes CVaR as a special case of $ProbR_{\Psi}$ and its corollary establishes the existence of the solution (novel result).

Numerical experiments demonstrate that hereby proposed version of PRT works on par with original version [Robey 2022] supported by pseudo code algorithm provided in Appendix to help adaptation.


**Strengths:**

+ Clarity: Accessible presentation of the problem, solution and use. Yet high standard of rigour is present in both main pair and appendices.
+ Theoretical framework with potential: Besides correcting the PRT and establishing existence guarantees (novel results) the paper develops clear mathematical toolkit for solving similar problems in adversarial robustness in rigour.
On top the Appendix A.6 PAC learnability for Lipschitz continuous Ψ - suggests future research and improvement of the statements of the paper.

+ Applicability/Impact: Appendix B1 contains the pseudo code for practical usability together with drawing the connections to CVaR as a surrogate objective.



**Weaknesses:**

- (minor) Numerical experiments presented are minimalistic yet arguably sufficient since theoretical guarantees have been provided. Intriguing results are mentioned but not furhterexplored or commented: “Note that the original or the geometric version of PRL should not be expected to match the adversarial robustness of classifiers trained with PGD attacks [Madry et al., 2017] or other worst-case optimization techniques. Instead, they shine with superior clean accuracies and easier training while maintaining probabilistic and a certain degree of adversarial robustness, as also observed by Robey et al. [2022].”

**Questions:**

The paper addresses the weak point of the PRT. Never the less, since an attack on such points would only lead to improved model (as also mentioned between lines 76-78), does it need to be addressed? Could you put more comments on why is the 'corrected version' of PRT needed? (Note that this is not questioning the added value of the other parts of the paper such as existence theorem and extensions)

**Limitations:**

Limitations have been commented on in a reasonable extend in Conclusions.

---

> ### Author Rebuttal · Authors · 2023-08-09
>
> Thank you very much for your kind words and for seeing merit in our paper! Below we provide point to point replies to your questions and remarks.
>
> - _Intriguing results are mentioned but not further explored or commented: "Note that ... or other worst-case optimization techniques."_
>   We will make this point clearer in the new version of the paper. Indeed, this is not a new observation but one that was already made by Robey et al. Since models trained with any variant of PRL actually never get to see adversarial attacks, but just samples from a certain distribution, their adversarial accuracies are typically much worse than for models trained with adversarial training. This is not surprising, since the PRL model just aims to enforce robustness on most of these samples, not on proper adversarial attacks.
>
> - _Nevertheless, since an attack on such points would only lead to an improved model (as also mentioned between lines 76-78), does it need to be addressed?_
>   It will not lead to an improvement at test time. Attacks that "correct" misclassified **training** data falsely lower the loss without actually improving classification on **test** data drawn from the same distribution. See the new figures and responses to Reviewers u3Qk and CsNz.

---

> > ### Comment · Reviewer_LMsa · 2023-08-15
> >
> > Thank to authors for their response. I have no further questions.

---

### Official Review · Reviewer_u3Qk · 2023-06-27

**Soundness:** 3 good
**Presentation:** 3 good
**Contribution:** 2 fair
**Rating:** 5
**Confidence:** 2

**Summary:**

The probabilistically robust learning (PRL) framework aims to trade-off clean accuracy and adversarial robustness in model training by introducing a loss that penalizes the suceptibility towards adversarially attacks in the perimeter of the data points. The authors of the paper under review note that the loss of this framework has some (possibly) unintended property: In binary classification, for misclassified data samples an adversarial attacker would attack the sample ‘correctly’, i.e. assign it the ground truth label instead of the wrongly learned one. For the PRL loss such a setting would decrease the loss, incentivising such behaviour. On noting this flaw, the authors propose an new loss which simply applies the PRL loss only to correctly classified points, and the standard ERM loss to the ones that are not (yet) correctly classified. The authors then go on to prove for various settings the existence of solutions to these minimization problems and finally show that empirically, their approach leads to a competitive clean accuracy while increasing the adversarial robustness for small datasets.

**Strengths:**

1. This work notes the question: What should one do with misclassified points in the context of adversarial robustness? Since they do not refer to previous work on this angle to adversarial robustness, I assume that this is a novel question. It seems like a very natural and relevant discussion point.
2. The existence results are novel and seem to be following a line of work which has previously explored the topic. Adding this new type seems relevant.
3. The writing is easy to follow and clear, as the authors successfully explain the differences between the different losses.


**Weaknesses:**

1. It is unclear and not discussed how detrimental considering misclassified examples in the PRL loss is in practice, possibly also depending on the quality and clean accuracy of the learned function. It would be especially interesting to discuss this as the large/small p results for MNIST and CIFAR seem to have opposite trends (improving/decreasing the adv score compared to “original” for large p).
2.  I believe the empirical results would be stronger, if datasets with a larger number of difficult datapoints were selected. The high accuracy in MNIST leaves only few datapoints to be misclassified and it is known that these may be ambigous even for human interpretation.
3. Weakness of the reviewer: I find myself unable to access correctness and relevance of the existence results.

**Questions:**

1. I think Fig. 1 could be improved. From looking at it immediately it was unclear to me if A is the set of ground truth green points, or the area in space which predicts a point to be green. I now understand it to be the latter, but I am still unsure whether the margin is actually stemming from a specific probabilistic model or if it is only illustrative (I think the latter again?). Since the penalization around the boundary stems from the training data points themselves, maybe the illustration would be more clear if one uses the [Robey 2022] way of putting the perimeter around the points and marking the penalized area, which was extremely helpful for my intuitive understanding when I checked that paper.  Note that red and green data sample points can be problematic to distinguish for the colorblind.
2. In line 55 $\mathcal{A}$ is not defined.
3. It seems that Eq. (4) is missing context which says that elements in A should have ground truth y=1.
4. The authors leave out the SVHN dataset from the previous work, which they otherwise compare with. I wonder if there is a specific reason for this.

**Limitations:**

I was mostly interested in the abstract due to the geometrical interpretation of decision boundaries and its relation to robustness.
Since I am more interested in the application side, for me personally the most intersting points of the paper are the observation that an adversary might correct a model and the definition and practical evaluation of the novel and improved loss.
I acknowledge that the relevance for the authors might weight more on the proof techniques, but I find myself not being able to assess their value and novelty with full justice.
Since the empirical evaluation and discussion of the idea of corrective adversaries within its context seem to be lacking, as described in Weaknesses, I assign the paper a score of 4.
However, I am willing to increase my score in line with the other reviewers if they are confident and convinced about the evaluation of the existing results.

*** Update after rebuttal: Edited Soundness score 1->3, updated overall score 4->5

---

> ### Author Rebuttal · Authors · 2023-08-09
>
> Thank you very much for your valuable comments on our paper. Below we provide point to point replies to your questions and remarks.
>
> - _Regarding the soundness score._
>   We realize the mathematics is a tad sophisticated. However, we stand by the correctness of our results and the validity of their proofs, as confirmed by the other reviewers. We hope you will revisit your score of 1 on soundness.
>
> - _It is unclear and not discussed how detrimental considering misclassified examples in the PRL loss is in practice._
>   Please also see our response to Reviewer CsNz. Although the empirical results do not clearly indicate improvement, we believe in the value of ML algorithms underpinned by strong and universal theoretical guarantees. Furthermore, since such examples can be easily created (see the new figure attached), we believe that sooner or later one will observe them also in "real data" settings.
>
> - _I think Figure 1 could be improved._
>   We have attached a new version which we hope is clearer. We also use different colors now. The crucial point is to illustrate that the original PRL loss function possesses pathological minimizers, which does not happen for the geometric modification we study in our paper.
>
> - _Leave out the SVHN dataset._
>   We left it out due to time constraints and computational resources. Preliminary results show this dataset tells the same story. We are happy to include it in the final paper.
>
> - _The observation that an adversary might correct a model and the definition and practical evaluation of the novel and improved loss._
>   This is really the core observation of our paper. As noted in our response to Reviewer j7Q6, we find it surprising that our novel modified loss does not detrimentally affect the practical evaluation, even as it affords stronger theoretical guarantees.
>
> - _I believe the empirical results would be stronger if datasets with a larger number of difficult datapoints were selected._
>   This is a good point and an excellent topic for future work. The main focus of the present work is on developing the theory, and regarding the experiments, we refer to the previous point.
>
> - _The relevance for the authors might weigh more on the proof techniques..._
>  As we mentioned in our author rebuttal, our main contribution is the geometric perspective we introduce to the probabilistically robust learning framework not just academic proofs.  From this perspective, we can explore the similarities between PRL and other instantiations of the idea of robust learning. Using our framework, we can establish that PRL and adversarial training are asymptotically (i.e., for small ε) alike, which we find quite surprising. This is Proposition 1 and the discussion thereafter. The tools we establish will enable other researchers to explore similar connections.

---

> > ### Comment · Reviewer_u3Qk · 2023-08-14
> >
> > I thank the authors for their reply.
> >
> > Considering the feedback you received from the other reviewers, I increased the score for soundness in line with them.
> >
> > Reviewer CsNz's thoughtful comment on the contribution is in line with my feeling, that the empirical evaluation lacks a good understanding of the actual number of difficult/misclassified points in a dataset that constitute the "flaw" from previous work. While I agree that such examples are easily constructed this does not imply that they actually occur/are problematic in real data or that we would expect to eventually see them in some application. I strongly agree therefore, that the empirical insight as is does not sufficiently support the theory and that instead of leaving it as a future direction, it would come very valuable to the reader as a package with the theoretical results. This should be acknowledged in the final draft.
> >
> > Additionally, I support reviewer CsNz's request to engaging slightly differently with the previous work, i.e. to consider your own work as an extension/parallel development to adress misclassified points rather than a specific flaw of the PRL framework.
> >
> > Overall, I do acknowledge that according to NeurIPS guidelines also purely theoretical works should be welcomed. This is why I raise my score to borderline accept - and ask the Area Chair to keep this in mind for their meta-review.

---

> > > ### Author Response · Authors · 2023-08-17
> > >
> > > We sincerely thank the reviewer for engaging in the review process.
> > >
> > > > *While I agree that such examples are easily constructed this does not imply that they actually occur/are problematic in real data or that we would expect to eventually see them in some application...*
> > >
> > > The frequency with which such pathological decision boundaries---and more generally a detailed study of the geometry of decision boundaries----is certainly an interesting research direction. We reiterate that we have no reason to expect to **not see them** in current or future applications. Our proposed modification to PRL is a straightforward and future-proof way to rule out such behaviour.
> > >
> > >  > *I strongly agree therefore, that the empirical insight as is does not sufficiently support the theory ... acknowledged in the final draft.*
> > >
> > > The goal of our experiments was not to make any claims regarding the frequency with which pathological boundaries occur. Instead, we aimed to verify that our proposed modification to PRL does not degrade it's empirical performance (as mentioned above). We are happy to make this clearer in the final draft.
> > >
> > > > *I support reviewer CsNz's request to engaging slightly differently with the previous work.*
> > >
> > > Agreed. We will reframe the relation of our work to the original PRL paper.

---

### Official Review · Reviewer_j7Q6 · 2023-07-06

**Soundness:** 3 good
**Presentation:** 3 good
**Contribution:** 3 good
**Rating:** 7
**Confidence:** 4

**Summary:**

This paper takes a fresh and geometric view on the method—Probabilistically Robust Learning (PRL) [Robey et al., 2022], an approach which interpolates between the robustness offered by adversarial training and the higher clean accuracy and faster training times of Empirical Risk Minimization. The paper proposes a geometric framework for understanding PRL, which identifying a subtle flaw in its original formulation and to introduce a family of probabilistic nonlocal perimeter functionals to address this. The paper proves existence of solutions using novel relaxation methods and study properties as well as local limits of the introduced perimeters.

**Strengths:**

The paper is well written and organized, and is innovative and original. The structure of the paper is clear and rigorous.

**Weaknesses:**

Experiments are not sufficient, only minist and cifar-10 datasets.

**Questions:**

1.Does the theoretical results in this paper hold for larger datasets?e.g.ImageNet
2.The proposed method can only satisfied the clean accuracy and adversarial robust trade-off under the PGD adversarial attack, what about other attacks,e.g. C&W attack.
3.In line 290,the results for Geometric on MNIST with p=0.1 in Table 1 is not consistent, for low clean accuracy and high robust accuracy compared with original.
4.small typo errors,line 292

**Limitations:**

It's better to evaluate the theoretical conclusions on larger datasets,for example ImageNet.

---

> ### Author Rebuttal · Authors · 2023-08-09
>
> Thank you very much for your kind words and for seeing merit in our paper! Below we provide point to point replies to your questions.
>
> - _Do the theoretical results hold for other attacks e.g. Carlini & Wagner, or for larger datasets?_
>   Yes, as our results are based on mathematical proofs and make few assumptions on the data or the adversary, they apply equally to all datasets and all adversarial attacks seeking to maximize a "misclassification loss" subject to a norm constraint. This includes the approach of Carlini & Wagner.
>
>   We did not evaluate on larger datasets as the emphasis of this paper is theory, and experimental results (datasets and attacks) are for comparison with the original PRL paper by Robey et al to show that our theoretical modification **does not detrimentally affect empirical performance**. Also, we have limited compute time.
>
> - _In line 290, the results for Geometric on MNIST with $p=0.1$ in Table 1 are not consistent, with low clean accuracy and high robust accuracy compared with the original._
>   The clean accuracies on the easy MNIST task are more or less the same (around 99%) for Geometric and Original, for all values of $p$ reported. The important thing to notice here is that the robust accuracies of the geometric method are better than those of the original one, significantly so for $p=0.1$.

---

> > ### Comment · Reviewer_j7Q6 · 2023-08-16
> >
> > Thank to authors for their reply,I have  no more questions.

---

### Author Rebuttal · Authors · 2023-08-09

We would first like to thank all reviewers for their valuable comments on our paper.

We would like to clarify the reasons why we believe our slight modification to the probabilistically robust learning (PRL) framework is meaningful. Therefore, before moving on to the individual responses to each reviewer's comments, we would like to use this global response to illustrate our main motivation in a simple setting:
Consider a dataset consisting of the two points $(x, \color{red} \text{red} \color{black}) $ and $(x', \color{blue} \text{blue}  \color{black})$ as shown in the attached pdf. Consider also the two binary classifiers $A$ and $\widetilde{A}$ in Figure 1 (a) and (b), respectively. Our argument in favor of our slight modification of the PRL model revolves around the following question:

**Question:**
Is $A$ a reasonable classification rule for the dataset $\{ (x, \color{red} \text{red} \color{black}) , (x', \color{blue} \text{blue}  \color{black}) \}$?

In our opinion, which we believe would be the opinion of most ML researchers, $\widetilde{A}$ should be strictly preferred over $A$ for the following reasons:
- Classifier $A$ does not even separate the two data points: both are assigned the same class label since both lie inside $A$ even though their true labels are different. Classifier $\widetilde{A}$, on the other hand, does classify them correctly.
- Classifier $A$ has a very curvy decision boundary and is consequently very susceptible to adversarial attacks. In short, $A$ is not robust.

Yet, note that, as illustrated in Figure 2 in the attached pdf, both classifiers $A$ and $\widetilde{A}$ are minimizers of the original PRL loss. This example can be scaled up to include arbitrarily many data points, revealing that this issue does not go away with a simple tuning of parameters. Instead, the issue is structural.

We notice that such "bad" classifiers may indeed arise when working with expressive families of classifiers such as deep neural networks (certainly these issues will not be present when working with linear classifiers). More fundamentally, ruling out "bad" classifiers as global optimizers of the PRL loss function in a general enough setting seems to be a very challenging problem that would have to be addressed by the original proponents of the PRL framework and requires finding *mathematical criteria* for the hypothesis class and the training data that prevent such behavior. To avoid this difficult theoretical undertaking, in our paper, we propose something simpler: we slightly modify the formulation of PRL so that these bad examples are structurally ruled out. Interestingly enough, our simple reformulation opens up the door to a wealth of geometric structure and conceptual insights that we believe are unexpected, novel, and solidly grounded in mathematical rigor.

We want to make the point that our main contribution, as already highlighted in the title, is the geometric perspective we introduce to PRL based on our modification. So, besides the modeling, what is important is not dry academic proofs (although we stand by our impression that these are interesting, elegant, and novel), but rather the questions and connections made between probabilistically robust learning and the geometry of decision boundaries.
We are interested in viewing PRL from a different perspective---a geometric one.
From this perspective, we can explore the similarities between PRL and other instantiations of the idea of robust learning.

---

### Decision · Program_Chairs · 2023-09-21

**Decision:**

Reject

**Comment:**

This paper (1) develops a geometric view on the framework of Probabilistically Robust Learning (PRL); (2) proposes a geometric framework for understanding PRL, and (3) introduces a family of probabilistic nonlocal perimeter functionals to address some of the limitations of PRL as noted int he paper.

The paper was discussed quite a bit among the reviewers (and myself). I thank the authors for their thorough responses and their effort to further inform the reviewers as well as myself about the contributions of the paper. In the end, some of the major concerns raised by the reviewers still remain. In particular, it is still unclear to most of the reviewers what the supposed "flaw" is (maybe flaw is not the right word here) and why this flaw an inherent limitation of PRL.  Moreover, some of the reviewers expressed concern about the empirical evaluations lacking to provided a good understanding of the actual number of difficult/misclassified points in a dataset which would point to the limitations of PRL explained in the paper.

Once the above issues are resolved, I believe that the updated paper will be an interesting contibution to the robust-ML community.